# Potentially Inappropriate Medications Involved in Drug–Drug Interactions in a Polish Population over 80 Years Old: An Observational, Cross-Sectional Study

**DOI:** 10.3390/ph17081026

**Published:** 2024-08-05

**Authors:** Emilia Błeszyńska-Marunowska, Kacper Jagiełło, Łukasz Wierucki, Marcin Renke, Tomasz Grodzicki, Zbigniew Kalarus, Tomasz Zdrojewski

**Affiliations:** 1Department of Occupational, Metabolic and Internal Diseases, Medical University of Gdańsk, ul. Powstania Styczniowego 9B, 81-516 Gdynia, Poland; marcin.renke@gumed.edu.pl; 2Department of Preventive Medicine and Education, Medical University of Gdańsk, 80-211 Gdańsk, Poland; kacper.jagiello@gumed.edu.pl (K.J.); lukasz.wierucki@gumed.edu.pl (Ł.W.); tz@gumed.edu.pl (T.Z.); 3Department of Internal Medicine and Gerontology, Jagiellonian University Medical College, 31-107 Kraków, Poland; tomasz.grodzicki@uj.edu.pl; 4Department of Cardiology, Congenital Heart Disease and Electrotherapy, Silesian Center for Heart Disease, 41-800 Zabrze, Poland; zbigniew.kalarus@kalmet.com.pl

**Keywords:** geriatrics, multimorbidity, polypharmacy, drug interactions, potentially inappropriate medications

## Abstract

The clinical context of drug interactions detected by automated analysis systems is particularly important in older patients with multimorbidities. We aimed to provide unique, up-to-date data on the prevalence of potentially inappropriate medications (PIMs) and drug–drug interactions (DDIs) in the Polish geriatric population over 80 years old and determine the frequency and the most common PIMs involved in DDIs. We analyzed all non-prescription and prescription drugs in a representative national group of 178 home-dwelling adults over 80 years old with excessive polypharmacy (≥10 drugs). The FORTA List was used to assess PIMs, and the Lexicomp^®^ Drug Interactions database was used for DDIs. DDIs were detected in 66.9% of the study group, whereas PIMs were detected in 94.4%. Verification of clinical indications for the use of substances involved in DDIs resulted in a reduction in the total number of DDIs by more than 1.5 times, as well as in a nearly 3-fold decrease in the number of interactions requiring therapy modification and drug combinations that should be strictly avoided. The most common PIMs involved in DDIs were painkillers, and drugs used in psychiatry and neurology. Special attention should be paid to DDIs with PIMs since they could increase their inappropriate character. The use of automated interaction analysis systems, while maintaining appropriate clinical criticism, can increase both chances for a good therapeutic effect and the safety of the elderly during treatment processes.

## 1. Introduction

The increase in life expectancy in the 21st century alongside the declining fertility rates has led to a progressive aging of the population. According to the World Health Organization’s report, the number of people over 65 years old is expected to increase from 524 million in 2010 (8% of the world’s population) to 1.5 billion (16% of the world’s population) by 2050 [1].

The main challenging factors in conducting safe and effective pharmacological therapy in older adults are changes in drug metabolism and elimination, as well as multimorbidity [2]. An additional disturbing aspect is the growing consumption of over-the-counter drugs (OTC), which are aggressively advertised and easily accessible [3].

Drug–drug interactions (DDIs) are reactions of at least two drugs that can lead to a quantitative and/or qualitative change in the action of one of them [4]. Potentially inappropriate medications (PIMs) can be defined as drugs for which use among older adults should be avoided due to the high risk of adverse reactions for this population and/or insufficient evidence of their benefits when safer and equally or more effective therapeutic alternatives are available [5]. The clinical context of drug–drug interactions detected by automated interaction analysis systems is exceptionally significant in seniors with multiple chronic diseases, who are in need of multidrug therapies according to guidelines of evidence-based medicine.

A reduction in inappropriate polypharmacy has been identified by the World Health Organization Third Global Patient Safety Challenge: Medication Without Harm as a major public health goal [6]. Properly conducted pharmacotherapy delivers multiple benefits such as a decreased risk of rehospitalization [7] and death [8] and cost reduction for the healthcare system [9], as well as an increased chance of achieving satisfactory therapeutic effects and improvement in life quality.

The aim of this study was to provide unique, up-to-date data on the prevalence of PIMs and DDIs in the Polish geriatric population over 80 years old. Furthermore, we aimed to determine the frequency and the most common PIMs involved in DDIs in our study group.

## 2. Results

In order to reflect the general population of Poland, the results were stratified according to the age structure of the Polish population over 80 years old in 2017. A detailed description of sampling and subsequent weighing can be found in the methodological publication [10]. 

### 2.1. Drug–Drug Interactions (DDIs)

DDIs were detected in 66.9% of the study group (n = 119) in a total number of 240 interactions. Detailed analysis showed that 60.7% of all respondents (n = 108) used drug combinations requiring treatment modification (Lexicomp category D; n = 197), while 17.4% of the entire group (n = 31) were treated with drug combinations that should be avoided (Lexicomp category X; n = 43). The mean number (95% CI) of DDIs per person was 1.35 (1.13–1.57), 1.11 (0.92–1.29) for category D interactions, and 0.24 (0.15–0.33) for category X interactions.

### 2.2. Potentially Inappropriate Medications (PIMs)

PIMs were detected in 94.4% of the study group (n = 168) in a total number of 473 substances. Detailed analysis showed that 84.8% of all respondents (n = 151) used substances with a questionable safety profile (FORTA class C; n = 317), while 62.9% of the entire group (n = 112) were treated with preparations that should be avoided in seniors (FORTA class D; n = 156). The mean number (95% CI) of PPIs per person was 2.66 (2.43–2.88), 1.78 (1.6–1.97) for class C substances, and 0.88 (0.74–1.01) for class D substances.

### 2.3. DDIs with PIMs

Verification of clinical indications for the use of substances involved in DDIs resulted in a reduction in the total number of DDIs by more than 1.5 times, as well as in a nearly 3-fold decrease in the number of interactions requiring therapy modification and drug combinations that should be strictly avoided.

However, despite that intervention, DDIs with PIMs were still found in 67 respondents (37.6% of all), with a mean number (95% CI) of 0.83 (0.56–1.10). Moreover, 40 respondents (22.5% of all) presented drug interactions requiring therapy monitoring (Lexicomp category D) due to the involvement of substances with a questionable safety profile (FORTA class C), while 4 people (2.5% of all) were treated with drug combinations that should be avoided (Lexicomp category X) with the involvement of substances contraindicated in seniors (FORTA class D). Detailed data are presented in Table 1. The qualitative analysis showed that the most common PIMs involved in DDIs were painkillers, and drugs used in psychiatry and neurology. Detailed data are presented in Table 2 and Table 3.

## 3. Discussion

Due to the significant increase in life expectancy that long-term care carries over to older age [11], a growing number of chronic comorbidities requires the implementation of multidrug regimens according to evidence-based medicine, including drugs with a narrow safety profile especially in seniors [12]. Reducing exposure to PIMs is associated with a lower risk of adverse drug reactions and hospitalization in older individuals; however, this has no influence on mortality [13]. Physician-led interventions are based on standardized tools, such as STOPP criteria (Screening Tool of Older persons’ Potentially Inappropriate Prescriptions) and START (Screening Tool to Alert doctors to the Right Treatment) [14], Beers’ criteria [15], the PRISCUS list [16], the MAI (Medication Appropriateness Index) [17], the Good Palliative–Geriatric Practice Algorithm [18] or the FORTA (Fit For The Aged) List [19]. The FORTA List stands out from other methods and is classified as a patient-in-focus listing approach, which requires complex medical knowledge about individuals. The clinical usefulness of the FORTA List has been validated in randomized controlled clinical trials [19].

The incidence of PIMs in the available literature varies from 1.2% in Norway reported by Fog et al. to 93.9% in Finland stated by Toivo et al. We observed a higher consumption of PIMs in the Polish geriatric population over 80 years old than in most studies [19,20]. Differences in PIM prevalence can be explained by the application of various measurement tools, as well as different study settings (home-dwelling vs. primary care vs. hospital) [21].

The reported incidence of potential DDIs in other studies also varies depending on the settings and patient sample [22,23,24]. According to the available literature, the highest DDI prevalence of 88.4% was reported in the Slovak Republic by Kolar et al. while the lowest prevalence at a level of 4.4% in Norway was reported by Fog et al. The prevalence of DDIs in our study was relatively high, which can be explained by the fact that our study group included the oldest seniors over 80 years old with excessive polypharmacy (≥10 drugs) who are at the highest risk of adverse drug reactions. In view of the gradual computerization of healthcare systems, clinical decision support programs are becoming more accessible. However, due to high sensitivity and low specificity, they can generate numerous alerts [25]. The available literature proves the effectiveness of decision support programs in deprescribing; however, in general, the interventions have little effect on hospital admissions or mortality [26].

Surprisingly, only a few studies involving a rather limited number of patients focused on the issue of DDIs occurring specifically with PIMs [21]. Furthermore, data concerning the oldest seniors are particularly sparse.

We have observed that the verification of clinical indications for the use of substances involved in DDIs resulted in a significant reduction in potentially clinically important DDIs. Nevertheless, DDIs with PIMs were still found in 37.6% of all respondents, with painkillers, and drugs used in psychiatry/neurology were the most common PIMs involved in DDIs. The few researchers who analyzed the topic reported similar benefits from such interventions [22,26,27,28,29].

To our knowledge, this is the first study in Poland and one of several studies in the world to present the important problem of DDIs with PIMs, with particular emphasis on the oldest people with excessive polypharmacy. The main limitation is the theoretical attempt to objectify the quality of pharmacotherapy with methods unable to fully reflect the complexity of clinical situations. 

## 4. Materials and Methods

### 4.1. Study Design

The study group consisted of participants from the nationwide, cross-sectional observational study NOMED-AF (NOninvasive Monitoring for Early Detection of Atrial Fibrillation), which included ECG monitoring, the completion of a detailed questionnaire, a follow-up survey, blood pressure measurements, and blood/urine sample collection. A detailed description of the methodology and sampling of the NOMED-AF study was presented in a separate publication [10]. All participants provided written informed consent prior to participation. The study was approved by the Independent Bioethics Committee for Scientific Research at the Medical University of Gdańsk (13/2020; 2020-04-21) and by the Bioethics Commission at the Silesian Medical Chamber in Katowice (26/2015; 2015-07-01). All research procedures were conducted in keeping with the Declaration of Helsinki and Good Clinical Practice.

### 4.2. Setting

The study was conducted from 2017 to 2018. Respondents were randomly selected by the Ministry of Digitization of the Republic of Poland based on a social security number database; therefore, they constituted a representative sample of the Polish population in terms of sex, age, and place of residence. Based on a detailed questionnaire, the data were obtained by a trained nurse directly from the respondent, their family, or caregivers, followed by the presentation of the packaging of all drugs. The interview covered all preparations (prescription drugs, over-the-counter drugs, vitamins, nutritional preparations, and dietary supplements) taken at least once in the two weeks preceding the study (including drug name, form, single dose, and dosing frequency). Respondents provided information on diagnosed chronic diseases and were asked to present discharge cards from previous hospitalizations. Based on these data, individuals were assigned codes from the International Classification of Diseases, Tenth Revision (ICD-10).

### 4.3. Participants and Sample Size

The specific inclusion criteria for this study were an agreement to provide information on taken drugs, using at least ten active substances (excessive polypharmacy, EPP), and age over 80 years. The study group comprised 178 respondents, including 79 women and 99 men. The mean (SD) age of the entire sample was 85.8 (4.2) years (85.5 [4.2] years for women and 86.1 [4.2] years for men). The calculated maximum error in the study group was 2%. Detailed analyses of comorbidities and sociodemographic factors of the study group were presented in a separate publication [30].

### 4.4. Variables

The analysis of drug interactions between active substances was performed using Lexicomp^®^ Drug Interactions by Wolters Kluwer Clinical Drug Information (www.wolterskluwer.com/en/solutions/lexicomp/, accessed on 10 March 2021), which enables a simultaneous analysis of 50 active substances. Detected interactions were classified into one of five categories: A = “no known interaction”; B = “no action required”; C = “monitor therapy”; D = “consider modifying therapy”; and X = “avoid combination”. Category D and X interactions were considered for further analysis as drug–drug interactions (DDIs) of potential clinical significance.

The analysis of potentially inappropriate medications was performed using the FORTA List (Fit For The Aged) [19]. For each respondent, active substances were assigned to one of four classes: A = “indicated”; B = “favorable”; C = “carefully”; and D = “do not apply”. Class C and D substances were considered for further analysis as potentially inappropriate medications (PIMs).

### 4.5. Statistical Methods

Post-stratification was used to adjust the sample structure against the Polish population in 2017. The results are presented as percentages, median values with first and third quartiles, and mean values with 95% confidence intervals (CIs). The analysis was performed using the statistical package R version 3.6.3 (R Foundation for Statistical Computing, Vienna, Austria) and SAS 9.4 TS Level 1M5 (SAS Institute, Inc., Cary, NC, USA).

## 5. Conclusions

Although it is difficult to avoid the limitations of theoretical considerations that are not able to reflect complex clinical situations, ideally PIMs and DDIs should always be reviewed in older patients. However, special attention should be paid to those DDIs occurring with PIMs since they could increase their inappropriate character. The use of automated interaction analysis systems, while maintaining appropriate clinical criticism, can increase the chances of achieving a therapeutic effect while increasing the safety of the elderly during the treatment process. Further studies would be necessary to investigate the potential negative outcomes of DDIs in PIMs on adverse drug reactions and their long-term consequences, such as hospitalization, morbidity, and mortality.

## Figures and Tables

**Table 1 pharmaceuticals-17-01026-t001:** PIMs involved in DDIs.

	Lexicomp D + X	Classification of DDIs	
		Lexicomp D	Lexicomp X
Mean number of DDIs (95% CI)	1.35 (1.13–1.57)	1.11 (0.92–1.29)	0.24 (0.15–0.33)
Number of DDIs (n)	240	197	43
Number of patients with DDIs (n)	119	108	31
Mean number of DDIs with PIMs (95% CI)			
Forta Class C + D	0083 (0.56–1.1)	0.69 (0.49–0.9)	0.14 (0.03–0.25)
Forta Class C	0.43 (0.28–0.59)	0.38 (0.24–0.52)	0.06 (0.01–0.1)
Forta Class D	0.40 (0.17–0.63)	0.32 (0.16–0.46)	0.08 (0–0.18)
Number of DDIs with PIMs (n)			
Forta Class C + D	148	123	25
Forta Class C	77	67	10
Forta Class D	71	56	15
Number of patients with DDIs with PIMs (n)			
Forta Class C + D	67	62	12
Forta Class C	44	40	8
Forta Class D	33	31	4

DDIs—drug–drug interactions; PIMs—potentially inappropriate medications.

**Table 2 pharmaceuticals-17-01026-t002:** Most common PIMs involved in DDIs from Lexicomp category D.

PIMs Involved in DDIs	Description	Solution
Painkillers		
NSAID–ASA	Increased risk of bleeding. Diminished cardioprotective effect of ASA. Decreased serum concentration of NSAIDs.	Monitor for increased risk of bleeding. Use alternative analgesics (e.g., acetaminophen).
NSAID–LD	Diminished diuretic effect of LD. Enhanced nephrotoxic effect of NSAIDs.	Monitor for decreased therapeutic effects of LD or evidence of acute kidney injury. Consider using an NSAID with lesser potential for interacting (e.g., diflunisal, flurbiprofen, ketoprofen, and ketorolac).
NSAID–VKA	Enhanced anticoagulant effect VKA. Increased risk of bleeding.	Monitor for increased risk of bleeding. Use alternative analgesics (e.g., acetaminophen).
NSAID–NSAID, e.g., Diclofenac–Aceclofenac; Diclofenac–Meloxicam; Diclofenac–Ketoprofen	Increased risk of some adverse effects of NSAIDs, including gastrointestinal adverse effects.	Monitor for increased risk of bleeding. Use alternative analgesics (e.g., acetaminophen).
Tramadol–H1-AH	Enhanced CNS depressant effect.	Avoid combination. If combined, limit the dosages and duration of each drug to the minimum possible while achieving the desired clinical effect.
Tramadol–BZD	Enhanced CNS depressant effect.	Avoid combination. If combined, limit the dosages and duration of each drug to the minimum possible while achieving the desired clinical effect.
Tramadol–Clonidine	Enhanced CNS depressant effect.	Avoid combination. If combined, limit the dosages and duration of each drug to the minimum possible while achieving the desired clinical effect.
Tramadol–Hydroxyzine	Enhanced CNS depressant effect.	Avoid combination. If combined, limit the dosages and duration of each drug to the minimum possible while achieving the desired clinical effect.
Tramadol–Tizanidine	Enhanced CNS depressant effect.	Avoid combination. If combined, limit the dosages and duration of each drug to the minimum possible while achieving the desired clinical effect.
Tramadol–Quetiapine	Enhanced CNS depressant effect.	Avoid combination. If combined, limit the dosages and duration of each drug to the minimum possible while achieving the desired clinical effect.
Tramadol–Mianserin	Enhanced CNS depressant effect.	Avoid combination. If combined, limit the dosages and duration of each drug to the minimum possible while achieving the desired clinical effect.
Tramadol–Z-drugs	Enhanced CNS depressant effect.	Avoid combination. If combined, limit the dosages and duration of each drug to the minimum possible while achieving the desired clinical effect.
Drugs in neurology and psychiatry		
Quetiapine–Ropinirole	Diminished therapeutic effect of dopamine agonist.	Consider using an alternative antipsychotic agent.
Quetiapine–Levodopa	Diminished therapeutic effect of dopamine agonist.	Consider using an alternative antipsychotic agent.
Carbamazepine–BB	Diminished therapeutic effect of beta-blockers.	Consider an alternative for one of the interacting drugs in order to avoid therapeutic failure.
Carbamazepine–HMGCRI	Diminished therapeutic effect of statins.	Consider an alternative for one of the interacting drugs in order to avoid therapeutic failure.
Carbamazepine–VKA	Decreased serum concentration of VKA	Monitor INR. Adjust the dose of VKA.
Carbamazepine–Quetiapine	Decreased serum concentration of Quetiapine. Increased serum concentration of Carbamazepine.	Adjust doses of both drugs. Monitor response.
Carbamazepine–DHP-CCB	Increased metabolism of DHP-CCBs.	Monitor for reduced therapeutic effects of DHP-CCBs, and adjust the dose. Consider alternatives to DHP-CCBs.
Ergotamine–Nebivolol	Enhanced vasoconstricting effect.	Monitor for evidence of excessive peripheral vasoconstriction. Consider alternative drugs.
Phenobarbital–Hydroxyzine	Enhanced CNS depressant effect.	Consider a decrease in barbiturate dose.
Phenobarbital–Z-drugs	Enhanced CNS depressant effect.	Avoid combination. If combined, limit the dosages and duration of each drug to the minimum possible while achieving the desired clinical effect.
Phenobarbital–HMGCRI	Diminished therapeutic effect of HMGCRIs.	Consider an alternative for one of the interacting drugs in order to avoid therapeutic failure.
Phenobarbital–Isosorbide mononitrate	Diminished therapeutic effect of Isosorbide.	Consider an alternative for one of the interacting drugs in order to avoid therapeutic failure.
Z-drugs–Hydroxyzine	Enhanced CNS depressant effect.	Avoid combination. If combined, limit the dosages and duration of each drug to the minimum possible while achieving the desired clinical effect.
Selegiline–Levodopa	Risk of hypertensive reaction.	Avoid combination.
Risperidone–Levodopa	Diminished therapeutic effect of dopamine agonist.	Consider using an alternative antipsychotic agent.
Others		
Clonidine–BB	Enhanced AV-blocking effect, sinus node dysfunction, rebound hypertensive effect.	Monitor heart rate and blood pressure.
SU–DPP-4I	Enhance hypoglycemic effect.	Consider a decrease in SU dose when initiating therapy with a DPP-4I and monitor patients for hypoglycemia.
MRA–Potassium	Risk of hyperkalemia.	Monitor serum potassium concentrations and for other evidence of hyperkalemia (e.g., muscular weakness, fatigue, arrhythmias, bradycardia).
NDHP-CCB–HMGCRI	Increased serum concentration of HMGCRI.	Consider using lower doses of statin, and monitor closely for signs of HMGCRI toxicity (e.g., myositis, rhabdomyolysis, hepatotoxicity). Fluvastatin, pravastatin, and rosuvastatin may be less affected.
Verapamil–Eplerenone	Increased serum concentration of Eplerenone	Adjust the dose of Eplerenone
Amiodarone–VKA	Increased serum concentration of VKA.	Monitor INR. Adjust the dose of VKA.
Ciprofibrate–HMGCRI	Enhance adverse/toxic effect of HMGCRI.	Monitor for signs/symptoms of muscle toxicity. Consider using alternative drugs.
Theophylline–BZD	Diminished therapeutic effect of BZD.	Monitor the effect of BZD.
Theophylline–Non-selective BB	Diminished bronchodilatory effect.	Monitor for symptoms of reduced theophylline efficacy.
Dietary supplements		
Gingko biloba–ASA	Increased risk of bleeding.	Monitor for signs and symptoms of bleeding (especially intracranial bleeding). Consider using alternative drugs.
Gingko biloba–Turmeric	Increased risk of bleeding.	Monitor for adverse effects (e.g., bleeding, bruising, altered mental status due to CNS bleeds).
Gingko biloba–Piracetam	Increased risk of bleeding.	Monitor for adverse effects (e.g., bleeding, bruising, altered mental status due to CNS bleeds).
Gingko biloba–NSAID	Increased risk of bleeding.	Monitor for adverse effects (e.g., bleeding, bruising, altered mental status due to CNS bleeds).

ASA—acetylsalicylic acid; BB—beta-blocker; BZD—benzodiazepine; DDIs—drug–drug interactions; DHP-CCB—dihydropyridine calcium channel blocker; DPP-4I—DPP4 inhibitor; H1-AH—H1-antihistamines; HMGCRI—HMG-CoA Reductase Inhibitor; LD—loop diuretic; MRA—aldosterone receptor antagonist; NDHP-CCB—non-dihydropyridine calcium channel blocker; NSAID—non-steroidal anti-inflammatory drug; PIMs—potentially inappropriate medications; SU—sulfonylurea; VKA—vitamin K antagonist.

**Table 3 pharmaceuticals-17-01026-t003:** Most common PIMs involved in DDIs from Lexicomp^®^ category X.

PIMs Involved in DDIs	Description	Solution
Painkillers		
Nimesulide–NSAID	Enhanced adverse/toxic effect of NSAIDs.	Avoid combination.
Dexketoprofen–NSAID	Dexketoprofen may enhance the adverse/toxic effect of NSAIDs.	Avoid combination.
Drugs in neurology and psychiatry		
Quetiapine–Potassium	Enhanced ulcerogenic effect.	Avoid solid oral dosage forms of potassium chloride. Liquid or effervescent potassium preparations are possible alternatives.
Quetiapine–Sotalol	Enhanced QTc-prolonging effect.	Avoid combination.
Carbamazepine–Apixaban	Decrease serum concentration of Apixaban.	Avoid combination.
Clomipramine–Potassium	Enhanced ulcerogenic effect.	Avoid solid oral dosage forms of potassium chloride. Liquid or effervescent potassium preparations are possible alternatives.
Risperidone–Ipratropium	Enhanced anticholinergic effect.	Avoid combination. If not possible, monitor for evidence of anticholinergic-related toxicities.
Others		
Spironolactone–Amiloride	Risk of hyperkalemia.	Avoid combination.
Tolterodine–Potassium	Enhanced ulcerogenic effect.	Avoid solid oral dosage forms of potassium chloride. Liquid or effervescent potassium preparations are possible alternatives.
Doxazosin–Tamsulosin	Risk of hypotension and syncope.	Avoid combination.

DDIs—drug–drug interactions; NSAID—non-steroidal anti-inflammatory drug; PIMs—potentially inappropriate medications.

## Data Availability

The data are contained within the article.

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
