# Peer review of "Potentially Inappropriate Medications Involved in Drug–Drug Interactions in a Polish Population over 80 Years Old: An Observational, Cross-Sectional Study"

_pharmaceuticals, 2024, doi:10.3390/ph17081026_

Round 1

Reviewer 1 Report

Comments and Suggestions for Authors

In this manuscript, a study on the prevalence of Potentially Inappropriate Medications (PIMs) and Drug-Drug Interactions (DDIs) among the Polish geriatric population over 80 years old was conducted. The manuscript presents unique and up-to-date data on the clinical context of drug interactions detected by automated analysis systems in older patients with multimorbidities. The results have significant implications for improving pharmacotherapy quality and safety in elderly patients. The study was well-conducted, and the data provided are valuable for clinicians and researchers focusing on geriatric pharmacotherapy. However, there are several minor issues that need to be addressed before the manuscript can be considered for acceptance in the journal Pharmaceuticals:

1. The discussion should include a comparison with similar studies conducted in other countries to provide a broader context for the findings.

2. The authors should address the limitations of their study, including potential biases in the selection of participants and the representativeness of the sample.

3. Throughout the manuscript, minor grammatical and typographical errors should be corrected to enhance readability and professionalism.

4. Figures and tables should be checked for consistency in terms of units, labels, and legends. Ensure that all abbreviations are defined at their first appearance in the text.

Comments on the Quality of English Language

Minor grammatical and typographical errors should be corrected to enhance readability and professionalism

Author Response

Dear Sir or Madam,

thank You very much for taking the time to review this manuscript. Please see the attachment for the detailed responses and the corresponding revisions/corrections highlighted/in track changes in the re-submitted files.

Yours sincerely,

Emilia Błeszyńska-Marunowska, MD, PhD

Reviewer 2 Report

Comments and Suggestions for Authors

This manuscript aims to provide current data on the frequency and types of PIMs and DDIs in the Polish population over 80 years old. It employs the FORTA List for assessing PIMs and the Lexicomp® Drug Interactions database to identify DDIs. The study involves 178 home-dwelling elderly individuals, analyzing their prescription and non-prescription drug use. Findings indicate that 66.9% of participants had DDIs, and 94.4% had at least one PIM. The study highlights the necessity for automated interaction analysis systems to improve medication safety and therapeutic outcomes in elderly patients.

The topic is very interesting and important, and the authors have put in a lot of detailed work and effort. However, there are some key issues I would like to discuss with the authors.

Major points

  1. The study's sample size is relatively small, with only 178 participants, and is confined to the elderly population over 80 years old in Poland. which may limit the generalizability of the results and mean the results are not applicable to elderly populations in other countries or regions. A larger sample size may enhance the reliability and external validity of the findings. However, obtaining data sets from a larger population may present certain challenges. The authors must have substantial experience and insights in this area. I would like to know how the authors view this issue, and what are the difficulties in increasing the sample size for this study.
  2. While identifying prevalent PIMs and DDIs, the study lacks an in-depth analysis of the underlying mechanisms and specific clinical implications of these interactions. Understanding the mechanisms behind specific PIMs and DDIs is essential for developing targeted interventions and guidelines for safer medication use. This limitation is important as it hampers the translation of findings into actionable clinical practices.
  3. The study relies on self-reported medication use without direct verification from medical records or pharmacy databases, which could affect data accuracy.
  4. The study does not track long-term health outcomes such as hospitalization rates, morbidity, or mortality associated with PIMs and DDIs.

Minor issues

  1. The full names of terms were not provided the first time their abbreviations appeared in the text, e.g., PIMs, DDIs.
  2. There is no need to put a period in the title.

In conclusion, while the study provides valuable insights into the prevalence of PIMs and DDIs in the elderly Polish population, addressing these issues in future research will enhance the robustness, applicability of the findings.

Comments on the Quality of English Language

The quality of the English language in the article is generally good.

Author Response

(The authors gave the same response as above.)

Reviewer 3 Report

Comments and Suggestions for Authors

1. Study on Potentially inappropriate medications and drug -drug interaction is scientifically sound ,but need to be corrected as follows

 - In Abstract section Author should give the full version of the words instead of abbreviations (e.g.- PIMs , DDIs etc.)

2. Ethical considerations

    -Sample size -178 is very small to come to any fruitful conclusion 

    -How did Author calculated the sample size for this study ?

    - Inclusion criteria did not included the full criteria e.g.  Education standard of patents , Marital status ,living conditions , number of hospitalizations in previous 5 years. Comorbid states or diseases for which  the patients have been given treatment

 - Exclusion criteria is totally ignored -Not mentioning which co-morbid states or diseases have been excluded 

 -List of diseases for which the PIMs  have been given and lead to drug-drug interactions have not been given

-Medications prescribed may be analyzed according to Beers criteria ( J. Am. Geriatr. Soc.  2015 ; 63:2227-46) 

3. References : Latest references should have been included

Author Response

(The authors gave the same response as above.)
